# Gender Differences in Diagnosis, Prevention, and Treatment of Cardiotoxicity in Cardio-Oncology

**DOI:** 10.3390/jcm11175167

**Published:** 2022-09-01

**Authors:** Shawn Simek, Brian Lue, Anjali Rao, Goutham Ravipati, Srilakshmi Vallabhaneni, Kathleen Zhang, Vlad G. Zaha, Alvin Chandra

**Affiliations:** 1Department of Internal Medicine, UT Southwestern Medical Center, Dallas, TX 75390, USA; 2Division of Cardiology, Department of Internal Medicine, UT Southwestern Medical Center, Dallas, TX 75390, USA; 3UT Southwestern Medical School, UT Southwestern Medical Center, Dallas, TX 75390, USA; 4Cardio-Oncology Program, Harold C. Simmons Comprehensive Cancer Center, UT Southwestern Medical Center, Dallas, TX 75390, USA

**Keywords:** cardiotoxicity, cardio-oncology, gender, sex, cancer, cardiovascular disease

## Abstract

Gender differences exist throughout the medical field and significant progress has been made in understanding the effects of gender in many aspects of healthcare. The field of cardio-oncology is diverse and dynamic with new oncologic and cardiovascular therapies approved each year; however, there is limited knowledge regarding the effects of gender within cardio-oncology, particularly the impact of gender on cardiotoxicities. The relationship between gender and cardio-oncology is unique in that gender likely affects not only the biological underpinnings of cancer susceptibility, but also the response to both oncologic and cardiovascular therapies. Furthermore, gender has significant socioeconomic and psychosocial implications which may impact cancer and cardiovascular risk factor profiles, cancer susceptibility, and the delivery of healthcare. In this review, we summarize the effects of gender on susceptibility of cancer, response to cardiovascular and cancer therapies, delivery of healthcare, and highlight the need for further gender specific studies regarding the cardiovascular effects of current and future oncological treatments.

## 1. Introduction

Great strides have been made in the field of medicine in elucidating the impact of gender on epidemiology, pathophysiology, clinical manifestations, disease progression, and treatment response [1]. This is particularly evident in the field of cardiology [2,3] and oncology [4], but there is a paucity of information of the effects of gender in the burgeoning field of cardio-oncology. In this review, we summarize the effect of gender on susceptibility of cancer, response to cardiovascular and cancer therapies, and delivery of healthcare, and we highlight the need for further gender-specific studies regarding cardiovascular effects of current oncological treatments. For the purposes of this review, we will primarily refer to the social constructs of male and female gender. Given the complexity of issues affecting and men and women with cancer, we will address not only biological differences between the sexes, but also psychosocial and healthcare/clinical factors that may have differential effects based on gender.

## 2. Gender Differences in Cancer Susceptibility

The lifetime probability of developing cancer is slightly higher in males (40.2%) when compared to females (38.5%) [5]. This discrepancy is even more apparent in childhood, where male children have about a 20% higher overall rate of incident cancer in comparison to female children [6]. The biological determinants of this observation are not well understood, and there are likely many factors involved. The obvious gender-specific hormonal factors as well as genetic and epigenetic differences certainly play a role in determining risk of cancer. Differences in environmental exposures between genders, such as sun exposure, diet, and tobacco use, have been implicated as well [6]. In fact, the current body of evidence suggests that environmental factors may play the predominant role in determining risk of cancer when compared to genetic factors. For example, a review of registry data from 44,788 pairs of twins in Swedish, Danish, and Finnish twin registries demonstrated that heritable genetic factors made only a minor contribution to the susceptibility of most malignancies [7]. These genetic factors have been described as effect modifiers, with the primary drivers of cancer risk being environmental [8]. Recently, differential response to oxidative stressors between males and females has been suggested in both animal models and humans [9]. With these factors in mind, we begin to understand the potential significance of differences in environmental exposures between genders.

Evidence also exists suggesting sexual dimorphism in immune surveillance. Furthermore, immune surveillance is now recognized as a major mechanism protecting hosts from cancer and slowing cancer progression [6]. Females are believed to mount more intense innate and adaptive immune responses in comparison to males. While this may contribute to the lower overall incidence of cancer in females, it likely also leads to the higher incidence of autoimmune diseases in women [6,10]. The mechanism of this phenomenon is thought to relate, in part, to the effects of sex hormones on the immune system. Sex hormone receptors are present on B and T lymphocytes, macrophages, and dendritic cells [6,10]. The effects of sex hormones, particularly estrogens, are thought to modulate the immune response, potentially leading to the differences seen between men and women [10]. An example of the interplay of the genetic, environmental, and immunologic factors can be found in human skin. Men are known to be more prone to skin malignancies [6]. Historically, increased sun exposure in males has been implicated; however, more recently, gender-specific differences in human skin have been increasingly recognized [11]. Ultraviolet radiation is known to induce immunosuppression in human skin, and this effect has been shown to be more significant in men [12]. This multifactorial model of cancer susceptibility is directly related to the gender differences observed in cancer risk.

## 3. Gender Differences in Response to Cancer Therapies

One of the most relevant gender differences to the clinical practice of cardio-oncology is the difference in response to cancer therapies. Understanding the risk of cardiotoxicity related to specific cancer treatment scenarios is critical to the field of cardio-oncology. Therefore, differential rates of cardiotoxicity by gender must be considered when managing patients with malignancies. These gender differences in response to cancer therapies include direct cardiotoxic effects as well as increased risk of subsequent cardiovascular events related to changes in hormone balance and development of known cardiovascular risk factors (hypertension, obesity, metabolic syndrome, etc.) [13,14,15,16,17].

Perhaps the most robustly studied gender difference in response to cancer therapy is the risk of cardiotoxicity after anthracycline use. Women are significantly more likely to develop cardiotoxicity in comparison to men when treated with anthracyclines, and this effect appears to be particularly prominent when the treatment occurs in childhood [13,14,15]. This observation has been documented in both early (<1 year) and late (>1 year) periods after anthracycline exposure. In a study population of 6493 children with cancer who received anthracycline therapy, cardiotoxicity was confirmed in 106 patients (1.6%) [13]. For the purposes of this investigation, cardiotoxicity was defined as congestive heart failure, abnormal measurements of cardiac function prompting discontinuation of therapy, or sudden death presumed to be cardiac in nature. The authors showed that the risk of cardiotoxicity was nearly two-fold higher (RR 1.9) in female patients compared to male patients [13]. Regarding late cardiotoxicity after anthracycline use, 120 children and adults who had received cumulative doses of 244 to 550 mg/m^2^ of doxorubicin were evaluated via echocardiography [14]. The participants were treated for either acute lymphoblastic leukemia or osteogenic sarcoma in childhood a mean of 8.1 years prior to the study [14]. Female participants were significantly more likely to show signs of decreased myocardial contractility when compared to males [14]. In fact, based on these observations, female sex is listed as an independent risk factor for anthracycline-related cardiotoxicity in the 2016 European Society of Cardiology Cardio-Oncology Practice Guidelines [18].

The mainstays of treatment for certain malignancies, such as breast and prostate cancers, include the use of hormonally active therapies. These, by design and often gender-specific use, lead to differential effects in men and women. Breast cancer is the most common non-cutaneous malignancy in women in the United States, and with modern therapies, survival rates for breast cancer are relatively high with >90% survival rate at 5 years [16]. This high survival rate can be attributed in part to hormonal therapies targeting estrogen receptor-positive breast cancers [16]. Both selective estrogen receptor modulators (SERMs), such as tamoxifen, and aromatase inhibitors, such as anastrozole, target this pathway, albeit in different ways. SERMs inhibit estrogen by interfering with estrogen binding to estrogen receptors [16]. Aromatase inhibitors lead to systemic depletion of estrogen levels by affecting the hypothalamic–pituitary feedback system [16].

Both aromatase inhibitors and SERMS have potential cardiovascular effects. There is evidence suggesting that aromatase inhibitor use increases cardiovascular events, particularly myocardial infarction, in comparison to placebo and tamoxifen [19,20]. A meta-analysis of seven randomized trials including 16,349 patients comparing anastrozole to placebo showed a modest trend toward increased cardiovascular events (OR 1.18, 95% CI = 1.00–1.40) [20]. A separate meta-analysis evaluated 19 randomized controlled trials including 62,345 patients treated with anastrozole versus tamoxifen. The authors demonstrated a statistically significant increase in cardiovascular events in those patients treated with anastrozole compared to those treated with tamoxifen (RR 1.19, 95% CI = 1.07–1.34), largely driven by myocardial infarction (RR 1.30, 95% CI = 1.11–1.53) [19]. In contrast to aromatase inhibitors, there are limited data suggesting SERMs are less likely to increase the risk of cardiovascular events and may even be protective [16]. SERMs have been shown to decrease low density lipoprotein cholesterol (LDL) and lipoprotein (a), while increasing the risk of diabetes and metabolic syndrome [16,21]. Further investigation is needed to better define the net effect of these changes; however, currently it appears the risk of cardiovascular events with SERMs is lower than with aromatase inhibitors [16,22]. Of note, the estrogen modulation effect of SERMs has been shown to increase the risk of venous thromboembolism and stroke [16,22,23]. As with any therapy, the potential risks of initiating a new treatment need to be considered with the benefits.

In women with premature surgical menopause after oophorectomy or those who require oral contraceptives or hormone replacement therapy as a result of cancer treatment, the risks of these interventions should also be considered gender differences within cardio-oncology. The debate continues regarding the risk/benefit ratio of hormone replacement therapy in menopausal women. There is an association from the Women’s Health Initiative suggesting increased risk of cardiovascular events with hormone replacement therapy, as well as other literature demonstrating increased risk of venous thromboembolism [24,25,26,27]. Furthermore, early-onset menopause appears to increase the risk of premature coronary artery disease and non-fatal cardiovascular events [28]. Specifically, regarding women with premature surgical menopause, including women with prior cancer, there is an association between premature surgical menopause and increased risk of incident cardiovascular disease [29]. In a study of 144,260 postmenopausal women in the United Kingdom, 644 (0.4%) had premature surgical menopause [29]. In comparison to women without premature menopause, those with premature surgical menopause had a significantly increased risk of incident cardiovascular disease (3.9 vs. 7.6%), and this association remained significant after adjustment for conventional cardiovascular risk factors and use of hormone replacement therapy (HR 1.87, 95% CI 1.36–2.58) [29].

Analogous to SERMs and aromatase inhibitors in breast cancer, androgen deprivation therapy has been used successfully in prostate cancer treatment and is associated with significant cardiovascular effects. There are four main classes of hormonally active anti-androgen therapies, including: surgical castration (orchiectomy), gonadotropin-releasing hormone (GnRH) agonists, GnRH antagonists, and androgen receptor antagonists. Surgical castration leads to rapid and sustained loss of testosterone. GnRH agonists initially increase testosterone levels and later lead to sustained reduction in testosterone by a negative feedback loop mechanism. Conversely, GnRH antagonists cause sustained testosterone reduction without the initial testosterone surge. Finally, androgen receptor antagonists are generally used in conjunction with GnRH agonists/antagonists and lead to further reduction in testosterone activity [16].

While there are limited data regarding surgical castration, the literature suggests increased risk of cardiovascular events with both GnRH agonists and antagonists [16]. A meta-analysis of observational data compared GnRH agonist use to no androgen deprivation therapy and showed an overall increased risk of cardiovascular death (HR 1.36, 95% CI 1.10–1.68) as well as increased risk of myocardial infarction (HR 1.20, 95% CI 1.05–1.38) [30]. In comparison to GnRH agonists, GnRH antagonists are a relatively newer therapy and have shown some promising results specifically regarding reduction in the adverse cardiovascular event profile seen with GnRH agonists [16,31]. Relugolix (GnRH antagonist) was compared to leuprolide (GnRH agonist) in a recent phase 3 clinical trial of 930 participants with advanced prostate cancer [31]. The authors demonstrated a 54% lower risk of major adverse cardiovascular events in patients treated with relugolix compared to leuprolide (HR 0.46, 95% CI 0.24–0.88) [31]. Despite this promising reduction in cardiovascular risk, the assessment and management of the risk of cardiovascular events in patients treated with androgen deprivation therapies will continue to be a vital aspect of cardio-oncology. These therapies are felt to increase cardiovascular risk at least partially through increased prevalence of known cardiovascular risk factors. Androgen deprivation therapy has been associated with increased rates of obesity, metabolic syndrome, hypercholesterolemia, and insulin resistance [16]. While further data are needed to understand if this risk is modifiable by aggressive risk factor management, surveillance for dyslipidemia and diabetes with appropriate initiation of indicated medical therapies is prudent.

Interestingly, a differential response to radiation therapy in men and women has also been documented. A recent meta-analysis of 10 observational studies evaluated rates of incident cardiovascular disease and cardiovascular mortality among 13,975 patients treated for Hodgkin’s lymphoma with radiation therapy [32]. The authors showed a significantly increased risk of incident cardiovascular disease and cardiovascular mortality in women compared to men after radiation therapy (OR 3.74, 95% CI 2.44–5.72) [32]. The authors chose this primary endpoint with the goal of evaluating events they thought would be related to the development of radiation-associated coronary artery disease. This finding may suggest a biological difference in the way men and women tolerate and recover from chest radiation. As with other medical therapies, a dose-related phenomenon has been proposed and may suggest that further investigation is needed to define the ideal dosing and delivery strategies for chest radiation in female patients.

As a therapeutic class, immune checkpoint inhibitors (ICIs) have recently emerged as a promising cancer therapy with increasingly diverse oncologic indications [33]. While ICI therapy has been a significant advance in the management of many malignancies, we are now aware of multiple immune-related adverse effects of ICIs, including cardiovascular toxicities [34]. By blocking immune system checkpoints, a pro-inflammatory state is created with the goal of combatting malignancy, although the risks of toxicity and inflammatory syndromes are increased as well [34,35]. Pre-clinical (murine) and cellular models with the checkpoint inhibitor ipilimumab (cytotoxic T-lymphocyte-associated antigen 4 (CTLA-4) inhibitor) have demonstrated cardiotoxic effects mediated by NLRP3/IL-1β and MyD88 [35]. These findings were associated with significantly decreased fractional shortening and radial strain in comparison to untreated mice [35]. The most well-known and documented cardiovascular effect of ICI use in humans is ICI myocarditis. ICI use has also been associated with pericarditis, vasculitis, and cardiac arrhythmias [33]. Given that widespread ICI use is a relatively recent phenomenon, data regarding gender differences in cardiovascular toxicities are limited. Furthermore, the majority of clinical trials include only a small minority of female participants [33,36]. For example, in a meta-analysis of 20 randomized controlled trials with 11,351 participants treated with ICIs, only 33% of participants were women [36]. In this setting, authors have evaluated retrospective data from the US Food and Drug Administration Adverse Event Reporting System database to evaluate potential risk factors of ICI myocarditis [37]. Zamami et al. evaluated 13,096 participants treated with ICI and identified 107 cases of ICI myocarditis [37]. Understanding the limitations of retrospective data, the authors did document increased risk of ICI myocarditis in women compared to men (OR 1.92; 95% CI 1.24–2.97) [37]. Interestingly, a preclinical murine model also suggests increased risk of ICI myocarditis in females compared to males [33,38]. Given these preclinical findings and associations noted retrospectively, future prospective randomized studies are warranted to better define gender-based risk of ICI myocarditis and other cardiovascular toxicities associated with ICI therapy.

The era of the COVID-19 pandemic has been an incredibly challenging period for everyone involved in healthcare. Regarding cardio-oncology specifically, we now know that patients with cancer and cardiovascular disease are at increased risk for developing severe COVID-19 with increased mortality rates [39,40,41]. Furthermore, it appears men, in general, are at increased risk of severe COVID-19 manifestations with increased mortality in comparison to women [42]. While the mechanism of this association is not fully understood, theories include increased prevalence of baseline cardiovascular disease and risk factors among men or gender differences in the immune response to COVID-19 infection. Interestingly, an association has been noted in men with prostate cancer and COVID-19 in relation to androgen levels [39,43]. Among 4532 male patients with laboratory confirmed COVID-19 infection, the risk of COVID-19 was higher in patients with a cancer diagnosis [43]. Specifically, among males with prostate cancer, the risk of developing COVID-19 was higher in males who were not treated with androgen deprivation therapy (ADT) compared to those who were treated with ADT (OR 4.05; 95% CI 1.55–10.59) [43]. The authors suggest that men with prostate cancer receiving ADT may be partially protected from COVID-19, indicating that high levels of circulating androgens may portend an increased risk of COVID-19 [39,43]. While these findings are thought provoking, further studies are needed to better understand potential gender differences in outcomes related to COVID-19 in patients with cancer and cardiovascular disease.

Finally, the field of cardio-oncology is a relatively young field in comparison to the larger fields of cardiology and oncology. With every new chemotherapeutic agent and class of chemotherapeutic agents, the field of cardio-oncology expands, and the rate of expansion has increased in recent years. This is one of the clinical challenges facing a practitioner in cardio-oncology. There are rarely robust long-term, gender-specific data regarding the cardiovascular effects and safety profiles of new chemotherapeutic agents when they first become available clinically. An understanding of the biologic pathways involved and strict monitoring/evaluation of adverse events is vital to building a gender-specific cardiovascular risk profile for each new oncologic therapy. As such, the field of cardio-oncology is constantly evolving. For example, ibrutinib, an inhibitor of Bruton’s tyrosine kinase, was FDA approved for mantle cell lymphoma initially and has since developed multiple expanded applications. With increased use clinically, we now know ibrutinib use is associated with increased risk of atrial fibrillation [44]. Furthermore, this risk appears to be greater in elderly male patients [44]. While the pathophysiology of ibrutinib-related atrial fibrillation is not fully understood, defining these gender-specific observations may help guide future studies to determine the underlying biologic mechanisms.

## 4. Gender Differences in Response to Cardiovascular Therapies

As we consider appropriate management for patients presenting with cardiotoxicity secondary to cancer treatment, the mainstays of therapy include medications used routinely for other non-cancer-related indications in cardiology. For example, guideline-directed medical therapy for heart failure is often used in patients with anthracycline-induced heart failure. Given this association, gender differences in the response and adverse effect profile of many cardiovascular therapies should be considered gender differences within the field of cardio-oncology as well. The list of gender biased responses to cardiovascular therapies is long and has been relatively well studied. A few examples may be found in therapies routinely used in the treatment of heart failure, coronary artery disease, and hypertension (all potential sequelae of cancer therapies). Beta blockers are often used in cardio-oncology, and interestingly, at standard dosing, the maximal serum concentration and area under the curve for beta blockers has been found to be up to 50% higher in women compared to men [45]. Furthermore, adverse reactions to beta blockers are significantly more common in CYP 2D6-dependent beta blockers in women (metoprolol, carvedilol, nebivolol, and propranolol) compared to men [45]. Cough is a common side effect reported by patients treated with angiotensin-converting enzyme (ACE) inhibitors [46]. Women were found to be two-fold more likely to report cough compared to men when treated with ACE inhibitors [46]. Increased bleeding risk has been demonstrated in women treated with antiplatelet agents in comparison to men [46]. Hyponatremia and hypokalemia related to thiazide diuretic use are both more common in women compared to men [45,46]. Peripheral edema is more commonly reported in women treated with calcium channel blockers when compared to men [27,46]. This list highlights a small portion of gender-specific issues facing the field of cardio-oncology when using cardiovascular therapies as treatment for cardiotoxicity and cardiovascular disease related to prior oncologic therapies. Future studies may elucidate gender differences in efficacy and adverse event profiles for promising new therapies in the field of cardio-oncology, such as SGLT-2 inhibitors, which have been shown to have beneficial effects on left ventricular function and myocardial fibrosis in mice treated with doxorubicin [47].

## 5. Biological Factors

The most readily apparent gender differences in cardio-oncology stem from the biological differences between men and women [48]. These biological differences are genetic/epigenetic in origin and center around hormonal and immunologic differences between sexes. This not only includes sex-specific malignancies, such as ovarian, uterine, prostate, and testicular, but also gender differences in cancer susceptibility and differential response to treatments, both oncologic and cardiovascular.

## 6. Clinical Factors

In addition to the biological bases of gender differences in cardiology and oncology, there are also differences seen between men and women on a societal level. Representation of men and women is historically inconsistent in clinical trials across multiple specialties. One study by Steinberg and colleagues suggested that women are less represented in clinical trials relative to disease burden, especially in cardiology and oncology [49]. This is particularly troubling given that the leading causes of death in women are cancer and cardiovascular disease [50]. Examination of trials in cardiovascular medications demonstrates a participation-to-prevalence ratio (PPR) of 0.8 in trials in heart failure, coronary artery disease, and acute coronary syndromes [51]. Gender bias also affects the care offered to patients by their providers, with some data suggesting women are less likely to receive advanced diagnostic and therapeutic interventions for the same indication as their male counterparts [52]. Moreover, FDA safety data suggests there may even be differences in drug metabolism in women. A recent review of the pharmacokinetic data for 86 FDA-approved medications was notable for elevated blood concentrations and longer elimination time in women in 88% (n = 76) of the included medications [53]. Interestingly, in the 59 medications with clinically identified adverse reactions, gender differences in pharmacokinetics predicted the direction of gender differences in the rates of adverse reactions in 88% (n = 52) of the included medications [53]. Moving forward, better representation of women in clinical trials may shed light on the important gender differences in the biology, pharmacokinetics, and adverse reaction profiles of therapies in the fields of cardiology and oncology.

## 7. Socioeconomic and Psychosocial Constructs

In addition to the biological and clinical factors thus far discussed, there also exist major differences between genders rooted in psychosocial and socioeconomic constructs. Traditional gender norms in western society and male-centered hierarchy have coalesced to form persistent gender biases in education, societal status, and healthcare [52]. A prospective cohort study of 9164 Americans older than 65 between the years 2002 and 2004 found women had fewer hospital stays and less access to preventative medical care compared to men. When adjusted for increased health needs and decreased economic access, these differences persisted and further revealed that women had fewer physician visits during the 2-year study period [54].

Explanations for these intrinsic biases in medicine are diverse, with much of them rooted in differences ascribed to traditional gender norms. Women are often expected to be more “feminine,” that is, more maternal and sensitive, whereas men are expected to be “masculine,” that is, aggressive and direct. Researchers hypothesize that communication differences between male and female patients help explain many of the gender differences seen in these studies. Men may be perceived as more direct and succinct with their complaints as well as potentially more willing to undergo invasive procedures. Conversely, it is argued that women may be less direct during medical interviews regarding symptoms and more willing to forego hospitalization to tend to caregiving responsibilities [54,55].

Yet, differences in communication styles likely present an inadequate explanation for the gender difference phenomenon. In studies involving paper cases and video simulations with controlled interaction between patient and physician, gender biases persisted, with women being asked fewer questions and receiving fewer diagnostic tests [56,57]. Results from these studies suggest alternative explanations, such as stereotyped expectations of the health of women and men by physicians, or the use of established statistical differences in gender to guide individual patient care. In the case of Arber et al., the notion that middle-aged men are the primary demographic affected by coronary heart disease helps explain why middle-aged women were asked fewer questions and received less diagnostic testing compared to men, when women would have potentially benefitted from further medical interviewing and testing [57].

Even among gender-specific cancers such as breast cancer, significant disparities exist when accounting for race and ethnicity. Indeed, even though new cases of breast cancer are similar among Black and White women, Black women are disproportionately more likely to die from breast cancer at any age and are saddled with increased morbidity [58]. This fact has been attributed to a wide range of socioeconomic and cultural explanations. For example, Black women are more likely to live in poverty and therefore have less access to healthcare. Additionally, poverty is associated with lower rates of education and lack of information on breast cancer prevention and early detection, therefore disproportionately harming Black women. Furthermore, cultural factors such as misconceptions of Black women on their own susceptibility to breast cancer and increased rates of distrust in the medical system have been posited as possible explanations of this disparity [58].

## 8. Conclusions

Much like in other areas of medicine, there are significant differences between genders in cardio-oncology, particularly in cardiovascular adverse effects associated with cancer therapy. These differences are likely due to a variety of factors, including biological, clinical, and psychosocial/socioeconomic factors. More studies are needed to further investigate these differences and to find ways to improve outcomes in both men and women.

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
