# Peer review of "Gender Differences in Diagnosis, Prevention, and Treatment of Cardiotoxicity in Cardio-Oncology"

_jcm, 2022, doi:10.3390/jcm11175167_

Round 1
Reviewer 1 Report
Manuscript titled " Gender Differences in Diagnosis, Prevention, and Treatment of Cardiotoxicity in Cardio-Oncology" describe the influence of gender differences in diagnosis and prevention of anticancer-related cardiotoxicity. The overall structure is of good quality, methoda are clear and figures are of good quality. References are updated in this field but still to be improved in some aspects. Specifically:
1. Authors should explain, in the era of COVID-19, how COVID-19 could influence anticancer drug-related cardiotoxicity ( cite doi: 10.3390/cancers12113316 )
2. A more appopriate analysis of diabetes and cardiovascular complications in cancer patients should be made and how gender differences could affect the risk in diabetic population. Moreover, authors should add an updated description of actually available antidiabetic drugs with cardioprotective properties like SGLT2i ( cite doi: 10.1186/s12933-021-01346-y. )
3. Authors should add a single paragraph on ICIs-related cardiovascular events and how gender differences should affect the cardiovascular risk. Moreover, a small description of known preclinical and clinical data on ICIs-induced cardiotoxicity should be made ( cita: doi: 10.3390/jpm10040179.)
Reviewer 2 Report
In this review article by Simek et al., “Gender Differences in Diagnosis, Prevention, and Treatment of Cardiotoxicity in Cardio-Oncology,” the author reviewed gender-based outcomes of cardiotoxicity during cancer prevention and treatment. The authors focused on cancer occurrence and treatment, summarized the causative factors, and discussed cardiac dysfunction caused by chemotherapeutic agents. Although this is a well-written review article, many more areas such as hypertension, myocardial fibrosis, atherosclerosis, and cardiac arrhythmia need to be discussed about the response in males and females during cancer treatment. Details about why men/women are more responsive than others remain exclusive.
Reviewer 3 Report
- In generally, the paragraphs are so long. They could be shorter to make it easier to read and understand, especially paragraphs that start in lines 118 and 165
- Line 52: These statistics are from reference 6, not from reference 5
- Line 94: reference?
- Line 101: This paper concluded that early clinical cardiotoxicity in children treated with anthracycline is rare (1,6%). So this paper is not important enough to demonstrate the difference of the cardiotoxicity between women and men, even though the authors showed that the risk of cardiotoxicity was nearly twofold higher female patients compared to male patients. Another study could better illustrate the female sex as a risk factor of cardiotoxicity.
- Line 156:¨... there is EVIDENCE of an association between premature surgical menopause and increased risk of incident cardiovascular disease¨ instead ¨...there is an association between premature surgical menopause and increased risk of incident cardiovascular disease¨
- Line 179: as is well established that androgen deprivation are associated with CVD risk factors, I suggest that this metanalisys should not be cited;
- Lines 213 to 221: ¨Finally, the field of cardio-oncology.....constantly envolving¨
I suggest these phrases should be after line 248 as a conclusion of Cancer Therapies
Round 2
Reviewer 2 Report
None